# A software tool and strategy for peptidoglycomics, the high-resolution analysis of bacterial peptidoglycans via LC-MS/MS
Marcel G. Alamán-Zárate [1,7], Brooks J. Rady [1,7], Raphael Ledermann[2], Neil Shephard [3], Caroline A. Evans [4], Mark J. Dickman [4], Robert D. Turner[3], Aline Rifflet[5], Ankur V. Patel[1], Ivo Gomperts Boneca [5], Philip S. Poole[2], Marshall Bern [6] & Stéphane Mesnage [1] ✉

Peptidoglycan is an essential component of the bacterial cell envelope—a mesh-like macromolecule that protects the bacterium from osmotic stress and its internal turgor pressure. The composition and architecture of peptidoglycan is heterogeneous and changes as bacteria grow, divide, and respond to their environment. Though peptidoglycan has long been studied via LC-MS/MS, the analysis of this data remains challenging as peptidoglycan's unusual composition and branching can't be handled by proteomics software. Here we describe user-friendly open-source tools and a web interface for building peptidoglycan databases, performing MS searches, and predicting the MS/MS fragmentation of muropeptides. We then use *Rhizobium leguminosarum* to describe a step-by-step strategy for the high-resolution analysis of peptidoglycan. The unprecedented detail of *R. leguminosarum*'s peptidoglycan composition (>250 muropeptides) reveals even the subtlest remodelling between growth conditions. These new and easier to use tools enable more systematic analyses of peptidoglycan dynamics.

Peptidoglycan (PG) is a ubiquitous and essential component of the bacterial cell envelope, which forms a single bag-shaped macromolecule (or sacculus) around the cell[1]. PG synthesis has been extensively studied since many antibiotics work by disrupting it, including widely used β-lactam antibiotics (like penicillin) and last resort antibiotics such as vancomycin[2,3]. The composition and remodelling dynamics of PG during growth, division, and differentiation can be critical for maintaining cell viability in response to changing environmental conditions. During this remodelling, PG fragments are naturally released into the environment; those released by the microbiota are important microbe-associated molecular patterns (MAMPs) recognised by the innate immune system[4]. They can contribute to acute or chronic inflammatory diseases and are thought to be key signalling molecules in the gut-brain axis[5,6]. PG fragments have also been shown to mediate more unusual symbiotic relationships, as in the case of the Hawaiian bobtail squid, where bioluminescent *Vibrio fischeri* provide the host with nocturnal

camouflage[7]. PG's unique role in bacterial adaptation, pathogenesis, and symbiosis makes it an essential molecule to study.

Whilst the overall structure of PG and its building blocks are well conserved, it is continually restructured and modified as bacteria grow and divide, introducing vast and often subtle complexity. Monitoring PG structural dynamics, therefore, requires automated, robust, and sensitive tools. Most analyses currently involve a biased identification of major peaks in UV absorbance chromatograms[8] that precludes the identification of low abundance or co-eluting muropeptides. The limited number of muropeptides commonly described this way (usually 10–25, even for so-called "high-resolution analyses")[9,10] does not provide enough detail to track the variation in important muropeptides like those corresponding to covalent protein anchoring. To achieve this greater level of detail, other studies have made use of LC-MS/MS—even proposing the term "peptidoglycomics" for the discipline in 2013[11]. Despite this, a lack of software tools and published search

[1]Molecular Microbiology, School of Biosciences, University of Sheffield, Sheffield, UK. [2]Department of Biology, University of Oxford, Oxford, UK. [3]Research Software Engineer team, University of Sheffield, Sheffield, UK. [4]Department of Chemical and Biological Engineering, ChELSI Institute, University of Sheffield, Sheffield, UK. [5]Institut Pasteur, Université Paris Cité, INSERM U1306, CNRS UMR6047, Biology and genetics of the bacterial cell wall Unit, Paris, France. [6]Protein Metrics, 225 Franklin Street, Boston, USA. [7]These authors contributed equally: Marcel G. Alamán-Zárate, Brooks J. Rady. ✉e-mail: s.mesnage@sheffield.ac.uk

strategies means that the LC-MS/MS analysis of PG has remained a tediously manual, error-prone, and inconsistent process.

To address this, several more comprehensive tools for peptidoglycomics have recently been developed[12–15]. Whilst these tools have vastly improved the consistency and throughput of LC-MS/MS analysis, they remain either inflexible, incomplete, or difficult to use.

Our previously described tool, PGFinder[15,16], focused on ease-of-use and the quantification of muropeptides in LC-MS datasets but left room for improvement. Here, we build on PGFinder in two key ways: (i) by improving the usability and capability of the existing MS tool, and (ii) by including new modules that automate additional analysis steps in the LC-MS/MS pipeline. Highlights include PGFinder's new, user-friendly web interface (https://mesnage-org.github.io/pgfinder/) and PGLang, a formal language for the concise description of muropeptides that enables both automated mass calculation and MS/MS fragment prediction. Finally, to demonstrate how these improvements fit into a complete analysis pipeline, we describe a step-by-step strategy that we use to characterise the changes in *Rhizobium leguminosarum's* PG composition when grown on minimal (as opposed to rich) media. Empowered by this approach, we report unprecedented PG complexity (>250 muropeptides) and accurately monitor subtle changes in the PG, laying the groundwork for more systemic analyses of muropeptide composition, cross-linking, and protein anchoring in the future.

## Results
### Enhancing PGFinder's existing functionality with an improved MS output and web interface
The first published version of PGFinder (v0.02[15];) offered automated MS analysis but search outputs still required post-processing in Excel to build a final table of muropeptides. Processing involved Δppm calculation and

consolidation of intensities (sum) across retention times. Now, in version 1.3.2, PGFinder automatically picks the best match according to its Δppm and consolidates search results into a table of muropeptides sorted by abundance. Finally, a new metadata column makes it possible to keep track of the data analysed, parameters, and PGFinder version used to generate each output. Taken together, these changes are a major step towards reducing the amount of manual processing required.

To use PGFinder v0.02 without installation, we previously provided an interactive Jupyter notebook hosted on MyBinder[17]. This made PGFinder significantly easier to set up and use than similar tools, but the resource limitations imposed by MyBinder regularly made loading our notebook slow (or even impossible). After loading, users also needed to ensure that all cells were run, in order, exactly once and needed to manually reset the notebook between each search. To circumvent these usability issues, we built a new, intuitive web interface that makes running an MS search as simple as uploading your deconvoluted data, picking a mass database, and clicking "Run Analysis" (Fig. 1). Since all computation is now done on the client-side (via WebAssembly), we no longer require hosting services like MyBinder and loading/computation times have been dramatically reduced. Moving to this interface has also allowed for bulk processing and made it trivial to add new modules like the Mass Calculator and Fragment Generator (Fig. 1). PGFinder is now easier to pick up than ever, further encouraging its adoption by others in the field.

### Condensing complex muropeptide structures into PGLang, a concise formal language
Before PGFinder could be expanded to handle tasks like mass calculation or MS/MS fragment prediction, we needed a way to model muropeptide chemistry in software. PG building blocks (muropeptides) are made of *N*-acetylglucosamine (GlcNAc) and *N*-acetylmuramic acid (MurNAc)

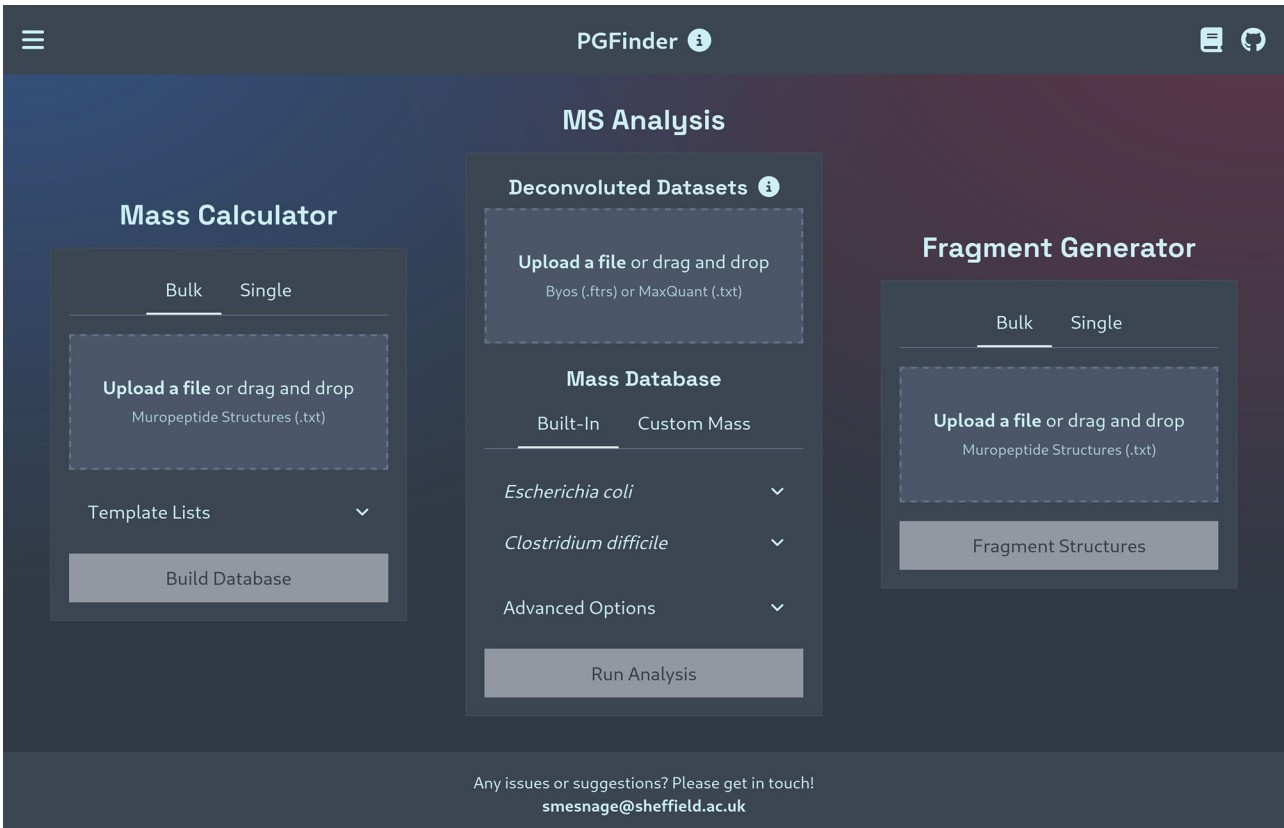

**Fig. 1 | PGFinder's web interface makes MS analysis easier than ever and enables new functionality.** The interface includes the original MS analysis module for identifying PG fragments from deconvoluted LC-MS data, as well as two newly developed modules: the mass calculator for building PGFinder-compatible mass databases, and the Fragment Generator for predicting MS/MS fragment ions.

disaccharides linked to a short (possibly branched) peptide stem containing both L and D-amino acids[1] (Fig. 2A). In PGFinder, these muropeptides are represented as monosaccharide and amino-acid residues that can be decorated with various modifications and bonded together to form a directed graph (Fig. 2B). Each residue contains distinct functional groups that are either free, modified, or donate/accept a particular bond (Fig. 2C). These rules ensure that every muropeptide is chemically valid, and tracking each muropeptide's free groups makes it possible to automatically identify potential modification sites and cross-linking positions.

To represent these muropeptide graphs compactly, we developed a language called PGLang with a minimal and straightforward syntax (Fig. 3). Each monomer is partitioned into a glycan chain (represented by lowercase letters) and a stem peptide (represented by uppercase letters). Lateral chains can be attached to any diamino or dicarboxylic amino acids using square brackets, and any residue can be modified using round brackets. Monomers can be connected via their glycan chain (~), or via cross-linked stem peptides (=). When monomers are connected via cross-linked peptides, the structure is followed by a bracketed list of cross-link descriptors: 3-3, 4–3, etc. A complete syntax diagram for PGLang is available in Fig. S1, and tables detailing the currently available monosaccharides, amino acids, and modifications are provided in Tables S1 and S2. Finally, to close the loop and move backwards from PGLang to a full molecular structure (including stereochemical information), we've included a PGLang to SMILES translator in PGFinder's new Mass Calculator module.

### Expanding PGFinder to automate the mass calculation and MS/MS fragment prediction

Once muropeptides described using PGLang have been translated into their chemical graph representations, implementing a number of new features becomes straightforward. Here that means automating two additional parts of the analysis pipeline that were previously out of scope for PGFinder: monoisotopic mass calculation and MS/MS fragment prediction. The mass of any given muropeptide is simply the sum of its residue, modification, and bond masses, and fragment prediction is a three-step process involving

bond cleavage, ion formation (depending on the bond broken, the acceptor and donor fragments may gain or lose a particular chemical group), and mass-charge ratio ($m/z$) calculation (all ions are currently $[M + H]^+$ adducts). Exposing this functionality are two new UI modules: the Mass Calculator, which generates mass databases that can be fed directly into the MS Analysis module, and the Fragment Generator, which produces a list of ions with PGLang-like descriptions that make it clear to users what fragment each ion represents. To get users started quickly, the Mass Calculator also includes several PGLang databases for common model organisms that can be easily downloaded and adapted using a text editor.

### *Rhizobium leguminosarum* as a model system for describing a five-step PG analysis strategy

The *R. leguminosarum* genome encodes many D,D and L,D-transpeptidases, so its PG structure is expected to be complex, making it a good model organism for testing our new PG analysis tools. To prepare some sample datasets, triplicate cultures of *R. leguminosarum* were grown in both minimal and rich media, and their PG was analysed via UHPLC-MS/MS (Fig. 4).

The chromatograms confirmed that *R. leguminosarum's* muropeptide profile was complex and revealed obvious differences between the two media conditions. Consequentially, the corresponding LC-MS/MS datasets were ideal for showcasing our comparative PG analysis strategy. Four sequential searches focus on monomers (step 1), modifications (step 2), PG-anchored proteins (step 3) and multimers (step 4) that inform a final, fifth search producing a comprehensive muropeptide quantification that can be used in statistical comparisons (Fig. 5).

**Step 1: Identifying PG monomers using MS and MS/MS.** An initial, unbiased search of the TY datasets was performed using PGFinder's "MS Analysis" module. The monomer database (DB_1; Table S3) contained 223 disaccharide peptides with stem lengths ranging from one to five amino acids. A total of 131 unique matches were found within 20 ppm of the observed masses; 78 of which were found in all three datasets, 22 in only two, and 31 in just one (Table S4 and File S1).

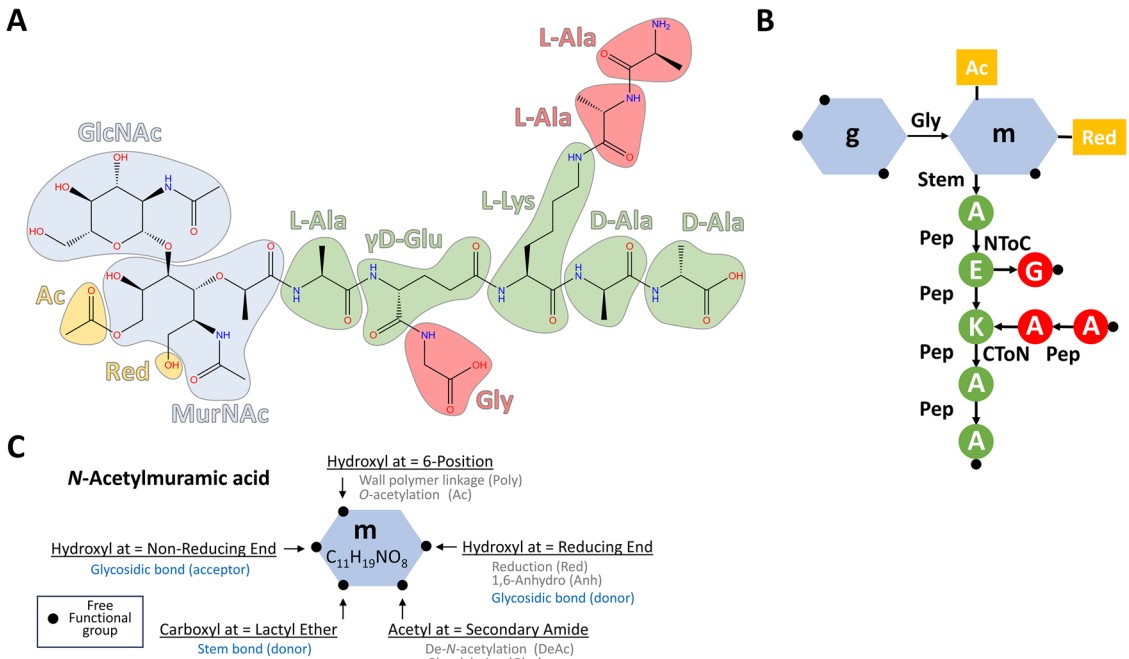

**Fig. 2 | Muropeptides are represented as a set of chemically linked, optionally modified residues. A** An example muropeptide showcasing several modifications (yellow) and peptide branches (red). **B** The same muropeptide converted to its chemical graph representation. Free functional groups (those that remain unmodified and unbonded) are shown as black dots on each residue, bonds are shown as labelled arrows pointing from donor to acceptor, and modifications are shown as yellow flags. **C** The five functional groups of *N*-Acetylmuramic acid are shown in detail, including the modifications each can have and the bonds they can donate/accept.

**Fig. 3 | PGLang is a simple language for describing potentially modified, branched, or cross-linked muropeptides.** Each cartoon PG fragment corresponds to a colour-coded PGLang structure. Monosaccharides forming glycan chains are represented by hexagons whilst amino acids are represented by circles. Modifications are shown as flags protruding from the residue they modify (Am, Amidation; Anh, anhydroMurNAc; Glyc, Glycolylation). J represents *meso*-diaminopimelic acid.

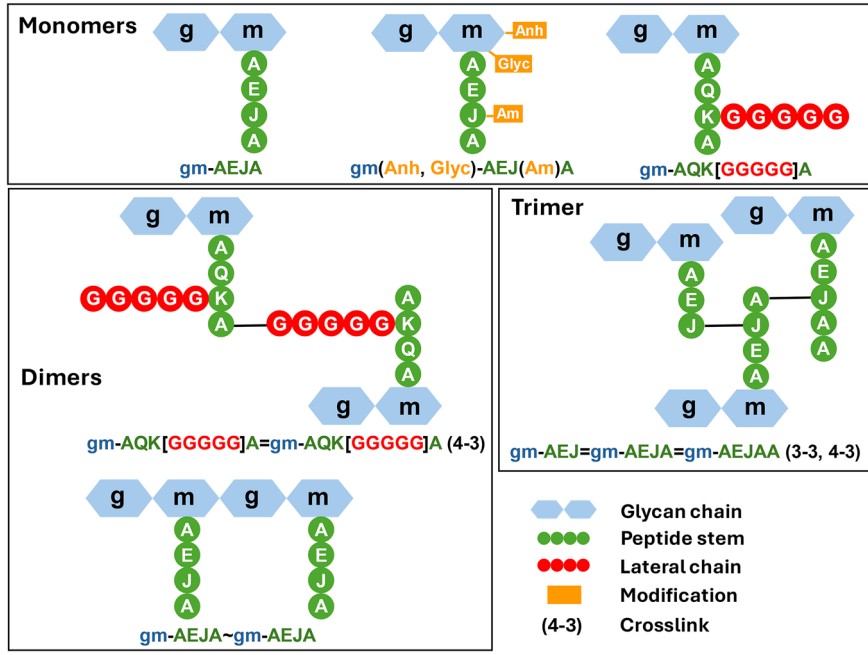

Several matches had very unusual compositions (e.g., gm-AEJCC or gm-AEJYS) and relatively high Δppm values (8.8 and 9.1, respectively), suggesting that these identifications resulted from mass coincidences. A manual inspection of MS spectra confirmed this hypothesis; many ions matching the theoretical *m/z* of these unusual muropeptides did not contain the signature ion corresponding to the loss of GlcNAc due to in-source fragmentation. To screen out these mass coincidences and resolve the structure of any isomers, we confirmed each monomer via MS/MS. We carried out a search of the TY datasets using the Byonic™ module of the Byos® software that can automatically analyse and score MS/MS spectra. The list of monomers to search for contained 423 muropeptides with stem lengths ranging from one to five amino acids, including disaccharide-tetrapeptides with all possible residues in position four (gm-AEJX) and disaccharide-pentapeptides with all possible combinations in positions four and five (gm-AEJXX; File S2). Based on the automatic scoring of MS/MS spectra, Byonic™ confirmed 39 monomers (Fig. S2). Further, manual inspection of the Byonic™ output allowed us to validate an additional 29 monomers that satisfied two criteria: they were fragmented in at least one replicate and contained at least half of the expected *b*-ions and half of the expected *y*-ions. The validated muropeptides contained many tetra- and pentapeptide stems with unusual residues in their final position—indicative of D,D or L,D-transpeptidase exchange activity. Two structures with identical masses, however, could not be differentiated with certainty (gm-AEJAG and gm-AEJQ, both 998.429165 Da). Seven monomers did not meet the criteria for validation (only one MS/MS spectrum across three biological replicates or less than half of the expected *b*- or *y*-ions). The lack of *b*- or *y*-ions for the three monomers (gm-AEJQE, gm-AEJEQ, and gm-AEJWW) prompted us to explore the corresponding MS/MS spectra and revealed that Byonic™'s automatic identification of these monomers was incorrect. The gm-AEJQE and gm-AEJEQ monomers were found to really be gm-AEJADA, whilst gm-AEJWW was, in fact, gm-AEJ = AEJ (3-3) (a dimer of tripeptides missing a disaccharide moiety). The 29 validated monomers that were present in all three TY replicates became the monomer database DB_2 (Table S5).

**Step 2: Identifying PG modifications.** Database DB_2 was then used to determine which monomers were the most abundant using PGFinder. A subset of 11 muropeptides, accounting for >90% of the monomer abundance (Table S6 and File S3), were selected to create a third database called DB_3 (Table S7). DB_3 was then used to search for six different modifications using PGFinder. The modifications considered are listed in Table 1 and can be sorted into glycan modifications (de-acetylation, *O*-acetylation, and 1,6-anhydroMurNAc), peptide modifications (amidation), and hydrolysis products resulting from Glucosaminidase (loss of GlcNAc) or amidase activity (presence of an extra GlcNAc-MurNAc). For each modification, matches were consolidated (summing the intensities of matches found at different retention times), and matches absent from any of the three replicates were discarded. Three additional criteria were then used to validate the modified muropeptide matches: (i) a retention time consistently higher or lower (depending on the modification considered) than the unmodified muropeptide; (ii) the presence of signature ions corresponding to each modification; (iii) a similar relative abundance of modified and unmodified muropeptides (Table 1).

As an example, Fig. 6A shows how nine putative AnhydroMurNAc-containing muropeptides were identified. Most modified muropeptides were present in all three replicates and had a consistently higher retention time than their unmodified counterparts. Next, we manually searched the MS/MS data for signature fragment ions predicted by PGFinder's "Fragment Generator" module. Fig. 6B summarises the MS/MS analysis of the gm(Anh)-AEJA monomer. Out of the 20 predicted fragment ions, 12 were present, including five out of nine possible signature ions (highlighted in red in Fig. 6B). The presence of these fragment ions ultimately contributes to validating the gm(Anh)-AEJA monomer.

On average, AnhydroMurNAc-modified muropeptides were 10% as abundant as their unmodified counterparts, with a particularly high proportion of the disaccharide-tripeptide gm-AEJ being modified (48% of its unmodified intensity) (Fig. 6A). In the end, only the anhydro versions of the three most abundant monomers (gm-AEJA, gm-AEJG, and gm-AEJ) could be confirmed (Fig. 6A), though this was in part due to a lack of MS/MS data for the other matches. Five other modifications (Table 1) were searched for using the same strategy, but none of these modifications could be confirmed via MS/MS (Fig. S3).

**Step 3: Identifying outer membrane proteins covalently anchored to the PG.** The covalent anchoring of outer membrane β-barrel proteins to the PG helps maintain cell envelope integrity in Alphaproteobacteria[18,19]. We identified ten putative β-barrel proteins encoded by the genome of *R. leguminosarum* bv. *viciae* (strain 3841) (Fig. S4) and investigated if any of them were anchored to the PG.

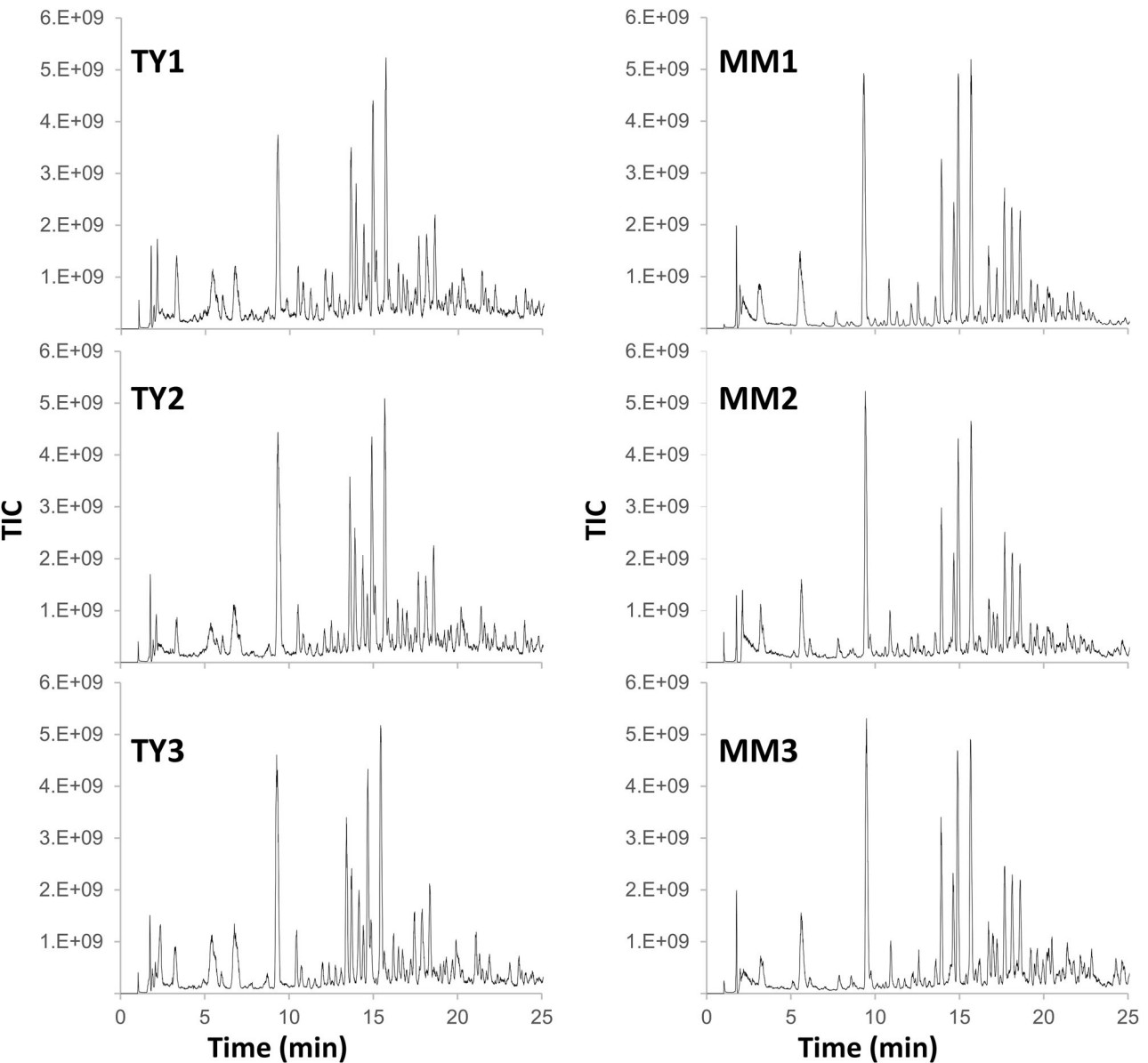

**Fig. 4 | *R. leguminosarum*'s PG composition is complex and varies between growth conditions.** Total ion chromatograms (TICs) show reduced *R. leguminosarum* muropeptides. PG was extracted from cells grown in either rich (TY) or minimal (MM) media. The TICs corresponding to each triplicate are shown.

A fourth database (DB_4, Table S8) was created to search for any amino acid scars left behind by β-barrel proteins that had been attached to the PG. Since tetrapeptide stems are thought to act as the donors during L,D-transpeptidase mediated protein anchoring, DB_4 contained muropeptides comprised of disaccharide tripeptides (gm-AEJ) followed by residues corresponding to the N-terminal of each anchored porin (with its signal peptide removed; Fig. S5A). Although trypsin digestion is expected to generate porin "scars" with a basic residue at the C-terminal, previous analyses revealed that muropeptides containing non-canonical and missed cleavages are common. To avoid missing any of these non-canonical "scars", the muropeptides in DB_4 contained every N-terminal porin sequence from 2 to 17 amino acids in length.

A total of 29 masses matching PG-anchored β-barrel proteins were found in all three TY replicates (File S4), though MS/MS data was only available to confirm muropeptides associated with RopA1,2,3, RopB, and pRL90069 (Fig. S5B). Collectively, muropeptides corresponding to these proteins accounted a total intensity of 3.51E + 08, representing 13.3% of all monomer intensity (2.64E + 09) (Table 2).

**Step 4: Identifying PG multimers and confirming their structure.** PGFinder's "MS Analysis" module was used to identify dimers and trimers resulting from both D,D- and L,D-transpeptidation using the monomers previously validated (DB_2; Table S6) as acceptors (apart from gm-AE). A total of 88 multimeric masses (41 dimers and 47 trimers) were present across all three replicates (File S5 and Table S9).

Out of the 41 dimers, 28 could be unambiguously classified as products of D,D or L,D-transpeptidation. This was possible because muropeptides like gm-AEJ=gm-AEJX (3-3) could only be formed via L,D-transpeptidation, and gm-AEJA=gm-AEJAX (4–3) structures could only be formed via D,D-transpeptidation. To differentiate the remaining 13 muropeptides, MS/MS analysis was necessary. They were either (i) isomers—with the same residues but distinct cross-linking; e.g., gm-AEJA=gm-AEJ (4–3) vs gm-AEJ=gm-AEJA (3-3)—or (ii) mass coincidences—with distinct compositions but the same chemical formula; e.g., gm-AEJ=gm-AEJQ (3-3) vs gm-AEJ=gm-AEJAG (3-3). In the Q vs AG case, MS/MS did not allow us to discriminate between the two structures, as a fragmentation between the C-terminal alanine and glycine was never observed. It was possible,

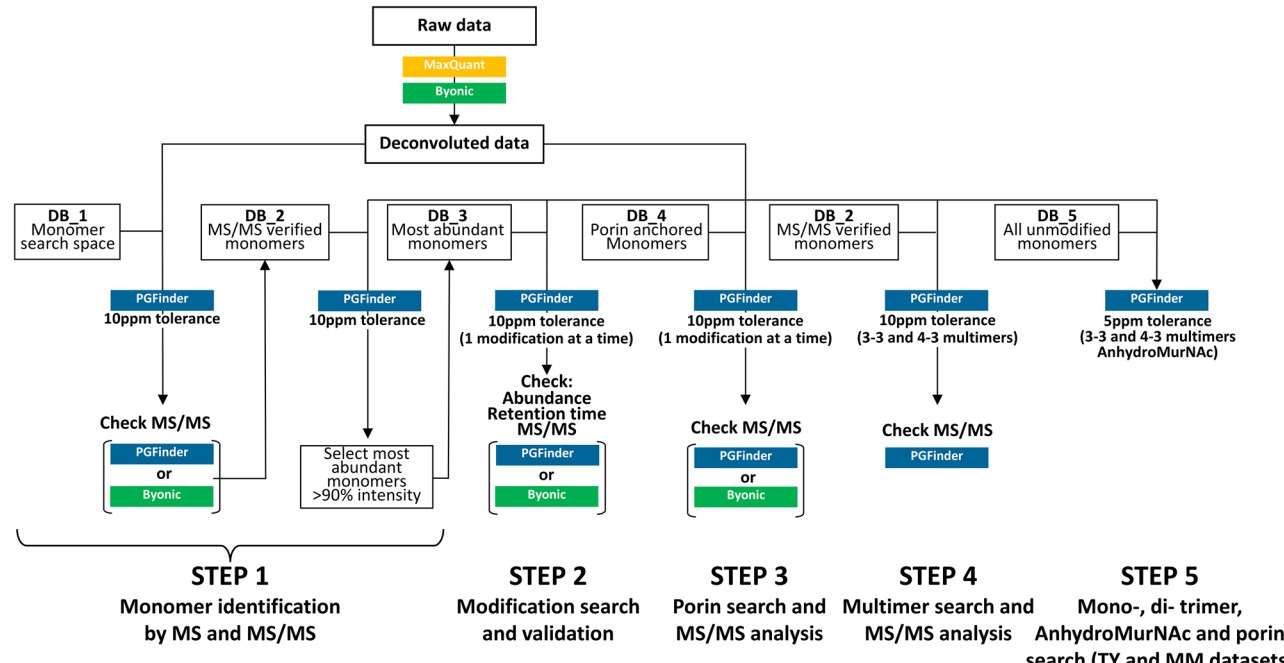

**Fig. 5 | An end-to-end strategy for PG structural analysis via LC-MS/MS.** Sequential searches were performed using PGFinder and Byonic™. The monomer database DB_2 was built based on MS/MS analysis. The most abundant monomers were then used to build DB_3, which was used to identify modified muropeptides. DB_4 contained muropeptides with a gm-AEJ stem followed by N-terminal porin sequences (with signal peptides removed). DB_2 was used to identify dimers and trimers, and then MS/MS data from matching output was manually inspected to differentiate between mass coincidences. A final search was carried out with DB_5, which combines muropeptides from DB_2 (monomers) and the muropeptides from DB_4 corresponding to the MS/MS confirmed RopA1,2,3, RopB, and pRL90069 porins. The final PGFinder search, with anhydroMurNAc modifications and 3–3/ 4–3 multimers enabled, was carried out with a 5 ppm tolerance. The final search output was manually inspected and modified to remove any known mass coincidences.

**Table 1 | A list of the PG modifications searched for, and the strategy used to validate them**

| Modification | | Description | Retention time | Mass change | Signature ion $m/z$ ([M+H]$^+$) | Relative abundance | Present |
|---|---|---|---|---|---|---|---|
| Glycan modifications | | | | | | 1) Similar % of modified monomers for all monomers | |
| AnhydroMurNAc | (Anh) | 1,6-anhydroMurNAc | Increased | −20.026 | 258.096 (AnhMurNAc) | 2) Relative abundance of modified monomers should be similar to the relative abundance of unmodified monomers | YES |
| Deacetylation | (DeAc) | Loss of acetyl group | Decreased | −42.010 | 162.077 (GlcN); 236.113 (MurN$^{red}$) | | NO |
| O-acetylation | (Ac) | Gain of acetyl group | Increased | +42.010 | 246.098 (GlcNAc+Ac) | | NO |
| | | | | | 320.134 (MurNAc$^{red}$+Ac) | | |
| Peptide modifications | | | | | | | |
| Amidation | (Am) | Glu or mDap amidation | Decreased | −0.984 | 172.109 (mDAP$_{NH2}$); 129.066 (Gln) | | NO |
| Hydrolysis products | | | | | | | |
| Loss of g | | Glucosaminidase | Decreased | −203.079 | 278.124 (MurNAc$^{red}$) | | NO |
| | | product | | | 206.087 (MurNAc$^{red}$ -Lactyl) | | |
| Extra gm | | Partial muramidase digestion | Increased | +478.180 | 276.108 (MurNAc) | | NO |

however, to assign one of the two possible isomeric structures—gm-AEJA=gm-AEJ (4–3) or gm-AEJ=gm-AEJA (3–3) to each of the three peaks in the extracted ion chromatogram ($m/z$ = 897.88; Fig. 7A).

As a first step, we predicted a list of fragment ions using PGFinder's "Fragment Generator" for each of the isomeric structures (71 fragments for the 3-3 dimer and 55 for the 4–3 dimer, coming together to form a set of 59 unique $m/z$ values; Table S10). As expected, a large proportion of the predicted ions (68%, 59%, and 58%) were detected in each peak. To assign each of the three peaks in the extracted ion chromatogram to a 3-3 or a 4–3 dimer,

we computed a list of signature ions for each structure (8 for the 4–3 dimer and 16 for the 3-3 dimer; Fig. 7B) and recorded their presence and intensity in each peak. The intensity associated with each set of signature ions allowed us to conclude that peak 1 was comprised of mostly 4–3 dimer whilst peaks 2 and 3 were mostly 3-3 dimer. The remaining ambiguous dimers were then analysed using the same strategy. Out of 46 total dimers identified, 21 contained 3-3 cross-links, and 25 contained 4–3 cross-links (Table S11).

Amongst the 47 trimers identified (Table S10), 26 matched only a single structure, Given the complexity of each MS/MS analysis and the prior

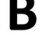

## A

| Structure [a] | | Intensity | % Anhydro | RT (min) | ΔT (min) [b] | Presence of signature ions |
|---|---|---|---|---|---|---|
| gm-AEJA\|1 | | 1.1E+09 | 3.5% | 9.50 ± 0.03 | + 6.62 | Yes; validated with 1+ and 2+ ions (TY1 dataset) |
| gm(Anh)-AEJA\|1 | | 3.7E+07 | | 16.12 ± 0.00 | | |
| gm-AEJG\|1 | | 2.4E+08 | 7.2% | 6.85 ± 0.04 | + 7.42 | Yes; validated with 1+ and 2+ ions (TY1 dataset) |
| gm(Anh)-AEJG\|1 | | 1.8E+07 | | 14.27 ± 0.01 | | |
| gm-AEJ\|1 | | 1.6E+08 | 48.0% | 5.48 ± 0.03 | + 8.10 | Yes; validated with 1+ and 2+ ions (TY1 dataset) |
| gm(Anh)-AEJ\|1 | | 7.9E+07 | | 13.58 ± 0.00 | | |
| gm-AEJAI\|1 | | 1.5E+08 | 0.6% | 21.40 ± 0.00 | + 6.99 | No fragmented ion available |
| gm(Anh)-AEJAI\|1 | | 8.8E+05 | | 28.39 ± 0.02 | | |
| gm-AEJF\|1 | | 1.5E+08 | 3.0% | 21.62 ± 0.00 | + 7.08 | No fragmented ion available |
| gm(Anm)-AEJF\|1 | | 4.5E+06 | | 28.70 ± 0.00 | | |
| gm-AEJAA\|1 | | 6.5E+07 | 0.9% | 10.86 ± 0.02 | + 6.40 | Only present in 2 of 3 datasets |
| gm(Anh)-AEJAA\|1 | | 5.9E+05 | | 17.26 ± 0.02 | | |
| gm-AEJK\|1 | | 4.6E+07 | 9.4% | 7.71 ± 0.06 | + 6.41 | No fragmented ion available |
| gm(Anh)-AEJK\|1 | | 4.3E+06 | | 14.12 ± 0.01 | | |
| gm-AEJAG\|1,  gm-AEJQ\|1 | | 4.6E+07 | 1.4% | 8.80 ± 0.05 | + 11.15 | No signature ions found. |
| gm(Anh)-AEJAG\|1,  gm(Anh)-AEJQ\|1 | | 6.5E+05 | | 19.95 ± 6.37 | | |
| gm-AEJAD\|1 | | 2.5E+07 | 11.7% | 9.61 ± 0.04 | + 6.21 | No fragmented ion available |
| gm(Anh)-AEJAD\|1 | | 2.9E+06 | | 15.82 ± 0.01 | | |
| gm-AEJAF\|1 | | 5.0E+07 | 0.0% | 15.4 ± 0.0 | - | No modification found |

[a] Non modified monomers are sorted by abundance (most abundant first)

[b] ΔT is defined as the difference (in min) between the average RT of the unmodified and modified muropeptide

## B

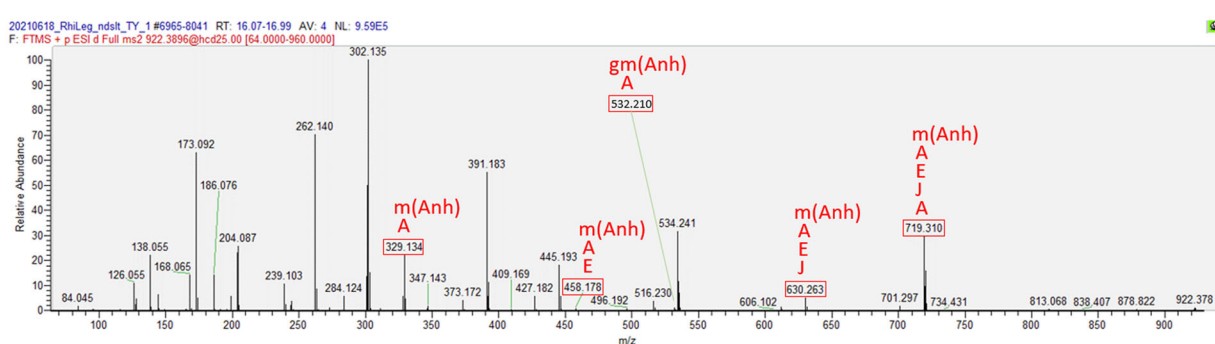

**Fig. 6 | The existence of AnhydroMurNAc-modified monomers can be confirmed via MS/MS. A** Summary of unmodified monomers and their anhydroMurNAc counterparts identified by PGFinder. For each muropeptide, the intensity and abundance of the AnhydroMurNAc modification are provided, alongside the retention time. The shift in retention time (ΔT (min)) and the presence of signature ions are indicated. **B** Example MS/MS spectrum showing the identified signature ions in red for a singly charged ($[M + H]^+$) ion corresponding to the gm(Anh)-AEJA muropeptide. RT retention time.

validation of dimers, we chose to simply assign ambiguous trimer matches to the structure built from the most abundant dimer linked to the most abundant donor. For example, structures like gm-AEJA=gm-AEJA=gm-AEJA (4–3, 4–3) were chosen over structures like gm-AEJA-gm-AEJ-gm-AEJAA (4–3, 3-3). The final list of 121 muropeptides, including monomers, dimers, porins, and their modified counterparts is described in Table S12.

**Step 5: Final quantification of muropeptides and comparison of growth conditions.** Growing *R. leguminosarum* in minimal media (MM) as opposed to rich TY media leads to changes in the muropeptide profile, suggesting that PG remodelling occurs under these conditions. We sought to apply the strategy described (and summarised in Fig. 5) to compare the PG structure of *R. leguminosarum* grown in rich and MM.

To perform a final quantification, we combined the monomers from DB_2 with DB_4's porin muropeptides from RopA1,2,3, RopB, and pRL90069 to generate the database DB_5 (Table S13). This database was then used to perform a "one off" search using PGFinder's new bulk processing feature. All three TY and MM datasets were searched with a low mass tolerance (5 ppm) and anhydroMurNAc modifications and 3-3/4–3 multimers enabled. Individual search outputs were consolidated and manually checked wherever retention times had a standard deviation of

more than 0.5 min. Dimer and trimer ambiguities were resolved using the strategy described in Step 4.

The final list of muropeptides contains 255 structures found across all three biological replicates of either condition: 65 monomers, 97 dimers, and 93 trimers. 111 muropeptides were exclusively found in the TY datasets, and 25 were exclusively found in MM (File S6). Comparing the two conditions reveals subtle differences in PG remodelling (Table 3 and Fig. 8). Growth in MM was associated with a significantly lower cross-linking index (28.5% ± 0.4% vs 31.3% ± 0.5%; $P = 0.002$) and a significant increase in glycan chain length (18.4 vs 21.1 residues; $P = 0.011$). Interestingly, we found a moderate but significant increase of 3-3 cross-links in the MM samples (64.7% ± 0.3% vs 62.6% ± 0.4%; $P = 0.003$). Whilst 3-3 cross-linking increased for dimers and trimers, L,D-transpeptidase-mediated exchange activity (which leads to non-canonical residues in the fourth position) drastically dropped in the MM samples: non-canonical AEJX peptide stems represented only 2.18% ± 0.4% of all muropeptides as compared to 26.6% ± 0.5% in the TY samples ($P > 0.001$). A significant decrease in the proportion non-canonical AEJAX peptide stems was also found in the MM samples (16.6% ± 0.3% vs 5.7% ± 0.4%; $P < 0.001$). Finally, a significant increase in the proportion of PG-bound porin peptides was observed in the MM datasets (5.2% ± 0.9% vs 9.1% ± 2.0%; $P = 0.037$).

**Table 2 | PGFinder identification of N-terminal peptides from RopA1,2,3, RopB, and pRL90069 anchored to the disaccharide-tripeptide gm-AEJ**

| | Structure | Av. Intensity | TY1 | TY2 | TY3 | Sum |
|---|---|---|---|---|---|---|
| RopA1,2,3, RopB or PRL90069 | gm-AEJAD | 2.52E+07 | 3.66E+07 | 2.05E+07 | 1.86E+07 | 2.52E+07 |
| RopA1,2,3 or RopB | gm-AEJADA | 1.69E+08 | 2.35E+08 | 1.66E+08 | 1.07E+08 | 1.69E+08 |
| RopA1,2,3 | gm-AEJADAIVA | 8.03E+07 | 7.96E+07 | 6.78E+07 | 9.36E+07 | |
| | gm-AEJADAIVAA | 1.14E+07 | 1.87E+07 | 9.81E+06 | 5.74E+06 | |
| | gm-AEJADAIVAAEPEPVE | 8.19E+06 | 9.59E+06 | 4.72E+06 | 1.03E+07 | |
| | gm-AEJADAIV | 7.24E+06 | 8.39E+06 | 7.55E+06 | 5.77E+06 | |
| | gm-AEJADAIVAAEPEPV | 2.41E+06 | 3.28E+06 | 1.14E+06 | 2.79E+06 | |
| | gm-AEJADAI | 1.24E+06 | 1.54E+06 | 1.14E+06 | 1.04E+06 | |
| | gm-AEJADAIVAAE | 9.13E+05 | 9.10E+05 | 9.26E+05 | 9.02E+05 | |
| | gm-AEJADAIVAAEPEP | 6.88E+04 | 6.88E+04 | ND | ND | 1.12E+08 |
| RopB or pRL90069 | gm-AEJADAV/gm-AEJADLG | 3.01E+06 | 3.91E+06 | 2.84E+06 | 2.28E+06 | 3.01E+06 |
| RopB | gm-AEJADAVDQVPEAPVAQ | 1.69E+07 | 1.44E+07 | 1.29E+07 | 2.35E+07 | |
| | gm-AEJADAVDQVPEAPVAQE | 2.50E+06 | 2.96E+06 | 2.31E+06 | 2.23E+06 | |
| | gm-AEJADAVD | 2.28E+06 | 3.18E+06 | 2.33E+06 | 1.32E+06 | |
| | gm-AEJADAVDQVPEAP | 1.40E+06 | 1.48E+06 | 1.61E+06 | 1.12E+06 | 2.31E+07 |
| pRL90069 | gm-AEJADLGTRTYEEPDLRNGV | 8.12E+06 | 1.56E+07 | 4.60E+06 | 4.17E+06 | |
| | gm-AEJADL | 5.55E+06 | 5.43E+06 | 6.16E+06 | 5.08E+06 | |
| | gm-AEJADLGTR | 5.13E+06 | 6.56E+06 | 4.55E+06 | 4.29E+06 | 1.88E+07 |
| | | | | | | 3.51E+08 |

**A**

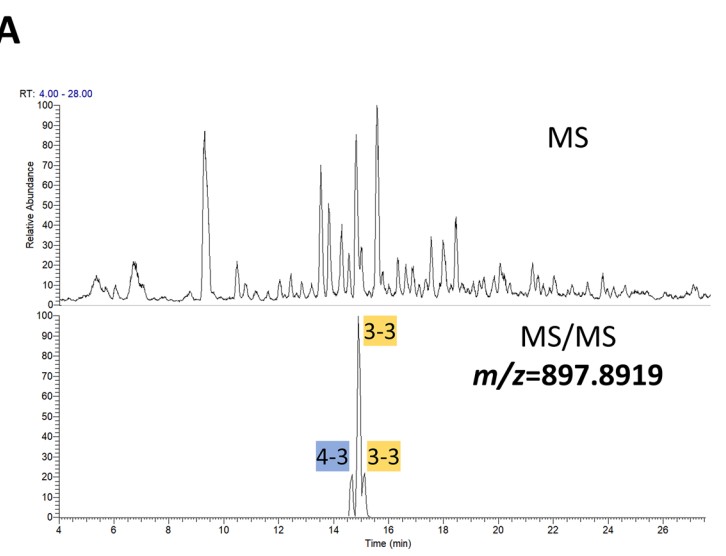

**B**

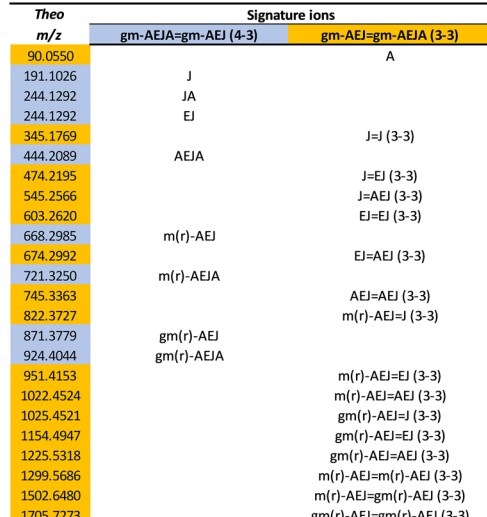

**Fig. 7 | Validation of gm-AEJ=gm-AEJA and gm-AEJA=gm-AEJ mass coincidence. A** Predicted list of signature ions for gm-AEJ=gm-AEJA (3–4 cross-link) and gm-AEJA=gm-AEJ (3-3 cross-link). **B** MS data corresponding to 897.8918 *m/z* ion (top panel) and extracted ion chromatogram showing 3 peaks at consecutive retention times (bottom).

## Discussion

The numerous functionality improvements to PGFinder's existing MS tool, as well as the new, PGLang-enabled mass calculation and MS/MS fragment prediction features, were key to our PG analysis strategy. By making these improved tools accessible through an easy-to-use web interface and laying out our approach in step-by-step tutorials, we hope to encourage others to adopt our rigorous and reproducible LC-MS/MS pipeline. Though PG structural analysis remains challenging, we feel that the improvements made to PGFinder throughout this work are a significant step towards the eventual elimination of labor-intensive and error-prone manual analysis.

Since PGFinder was first described, two other tools dedicated to the LC-MS analysis of PG have been published[13,14]. Every existing tool, PGFinder included, comes with its own trade-offs and rank differently when it comes to flexibility, completeness, and ease-of-use. HAMA, for example, is one of the more complete tools, covering the whole LC-MS/MS pipeline, but is written in MATLAB and lacks a GitHub repository. This makes it impossible to adapt or extend without a MATLAB licence and difficult to contribute those improvements back to HAMA. Additionally, HAMA doesn't currently build 3-3 cross-linked dimers, because they are often confused with 4–3 dimers due to the similarity of their fragmentation

**Table 3 | Comparative muropeptide analysis of PG extracted from cells grown in TY or MM**

| | TY Average SD | MM Average SD | Unpaired t test P value (significance) |
|---|---|---|---|
| Monomers (inc. porins) | 30.72% ±1.06% | 35.57% ±1.17% | |
| Dimers (inc. porins) | 49.65% ±0.85% | 42.94% ±0.31% | |
| Trimers (inc. porins) | 19.63% ±0.22% | 21.49% ±0.90% | |
| 3–3 | 62.56% ±0.45% | 64.68% ±0.31% | 0.003 (**) |
| 4–3 | 37.44% ±0.45% | 35.32% ±0.30% | 0.002 (**) |
| Cross-linking index | 31.30% ±0.49% | 28.53% ±0.43% | 0.002 (**) |
| AEJX monomers (excl. AEJA) | 4.86% ±0.21% | 1.56% ±0.34% | |
| AEJX dimers (excl. AEJA) | 16.01% ±0.50% | 0.50% ±0.06% | |
| AEJX trimers (excl. AEJA) | 5.70% ±0.05% | 0.12% ±0.01% | |
| All AEJX | 26.58% ±0.52% | 2.18% ±0.41% | <0.001 (***) |
| AEJAX monomers (excl. AEJAA) | 4.01% ±0.09% | 3.20% ±0.17% | |
| AEJAX dimers (excl. AEJAA) | 5.28% ±0.11% | 1.78% ±0.27% | |
| AEJAX trimers (excl. AEJAA) | 7.31% ±0.26% | 0.67% ±0.14% | |
| All AEJAX | 16.59% ±0.29% | 5.65% ±0.41% | <0.001 (***) |
| Anh monomers (inc. porins) | 2.18% ±0.08% | 2.26% ±0.15% | |
| Anh dimers (inc. porins) | 5.95% ±0.17% | 4.98% ±0.16% | |
| Anh trimers (inc. porins) | 0.89% ±0.02% | 0.07% ±0.02% | |
| All anhydro | 9.02% ±0.14% | 7.77% ±0.32% | |
| Chain length | 18.37±0.23 | 21.08±1.03 | 0.011 (*) |
| Porin monomers | 3.83% ±0.63% | 5.28% ±1.26% | |
| Porin dimers | 1.17% ±0.23% | 3.08% ±0.60% | |
| Porin trimers | 0.23% ±0.04% | 0.75% ±0.17% | |
| All porins | 5.23% ±0.87% | 9.12% ±2.01% | 0.037 (*) |

spectra[13]. In contrast, PGFinder is written entirely using open-source programming languages, follows software best practices, and is hosted on GitHub where anyone can easily contribute improvements back to the project. Though PGFinder doesn't yet automate MS/MS analysis, our signature ion approach makes it possible to know for certain if 3-3 or 4–3 cross-links are present, a critical distinction when it comes to assessing things like antibiotic resistance or L,D-transpeptidase activity. Another powerful tool for the in silico fragmentation of muropeptides, PGN_MS2, suffers from an incomplete description of its Python dependencies, making the installation process challenging. Additionally, by operating entirely at the atomic level, the fragment generation process is slow, and fragments can only be described in SMILES, making it difficult to tell at a glance which fragment came from which part of the muropeptide. Finally, though its fragment prediction is currently more complete than either HAMA or PGFinder's, it's limited to this task only; users will need to use another tool for the actual MS and MS/MS analysis[14]. PGFinder, on the other hand, requires no installation whatsoever, and its residue-graph abstraction makes generating fragments orders of magnitude faster than PGN_MS2 whilst giving them each useful PGLang-like names. Additionally, MS analysis can be done within PGFinder itself, only requiring additional software for data deconvolution/ feature extraction. The next steps for PGFinder are clear: more of the LC-MS/MS analysis pipeline can be covered by further automating tasks like cross-replicate consolidation, summary statistic generation, and MS/MS analysis/disambiguation. These changes, along with incorporating data deconvolution into PGFinder directly (eliminating the need for MaxQuant or Byos), would bring PGFinder closer to being a true, one-click muropeptide analysis tool.

We believe that the wider adoption of PGLang could help address the inconsistency of muropeptide descriptions throughout the literature. This inconsistency can make it difficult to understand the composition of many muropeptides; for example, the monomer GlcNAc-MurNAc-Ala-Glu-mDAP-Gly (gm-AEJG in PGLang) has been described in many ways: GM-Tripeptide + Gly[20], (NAG)(NAM)-AemG[14], AEmG[21], Tri-Gly[22], DS-TP-Gly[23], M3G[24], B-M-l(-A–E–H-G)[13], or even as numbers originally defined in other publications[25]. The description of dimers, trimers, and modifications are likewise inconsistent. By building on the existing intuition of those in the field, PGLang aims to remain intuitive whilst striking the right balance between concision and unambiguity. The automated monoisotopic mass calculation of PGLang structures will also help to address a surprising inconsistency in masses reported by the literature. For example, the theoretical monoisotopic mass of the major reduced monomer in *E. coli* (gm-AEJA, 941.407702 Da) is reported variably as: 941.4099 in ref. [26] ($\Delta ppm=2.3$), 941.4030 in ref.[21] ($\Delta ppm=5.0$), 941.41 in ref. [27] ($\Delta ppm=2.4$) or 941.4064 in ref. [28] ($\Delta ppm = 1.4$). Additionally, the PGLang to SMILES translator can be used to get stereoisomer-resolved structures that make obtaining a chemical formula, chemical drawing, or protein-ligand docking trivial (using a tool like Boltz-1[29];). Note that whilst PGN_MS2 does output SMILES structures for the muropeptides it fragments[14], these structures do not contain stereoisomer information and are therefore unsuitable for docking into stereospecific enzymes like L,D- and D,D-transpeptidases. By designing PGLang to be easy for humans to read and by including a number of useful tools for its translation and manipulation, we hope that it can become a standard nomenclature capable of improving consistency throughout the field.

Our proof-of-concept study describing PGFinder v0.02[15] was largely limited to a description of the software. The step-by-step strategy laid out in this paper allows any user with a basic understanding of PG structure to perform comprehensive structural analyses. Using *R. leguminosarum* as a model system, we identified 265 muropeptides, which represents (by far) the most comprehensive PG analysis to date and the first PG characterisation of this organism. This work provides a solid foundation for exploring cell envelope remodelling occurring throughout the rhizobial life cycle: from a free-living soil-dwelling bacterium to a terminally differentiated bacteroid that can fix atmospheric nitrogen. The remodelling of PG has been associated with morphogenetic changes during growth and exposure to various stressors, but how specific enzymes contribute to this adaptation remains poorly understood. The level of detail of our analysis will allow us to more easily investigate the roles played by these PG remodelling enzymes in the future.

Another aspect of PG analysis that requires highly sensitive tools is the description of covalently anchored proteins. In *R. leguminosarum*, we demonstrated that a large proportion of muropeptides contain N-terminal residues from β-barrel proteins. This covalent anchoring of β-barrel proteins is known to tether the outer membrane in the closely related genera *Coxiella* and *Brucella* and play a role in maintaining cell envelope integrity[18,19]. The increase in the proportion of PG fragments with β-barrel

**Fig. 8 | R. leguminosarum grown in minimal media (MM) has a different PG composition compared to those grown in rich media (TY).** The Sankey diagram shows the total PG composition broken down first by oligomerisation state, then by stem peptides. Branch size is proportional to percentage, and only peptides stems are represented. A, L- or D-alanine; E, γ-D-glutamic acid; J, *meso*-diaminopi-melic acid; X, any residue except Alanine.

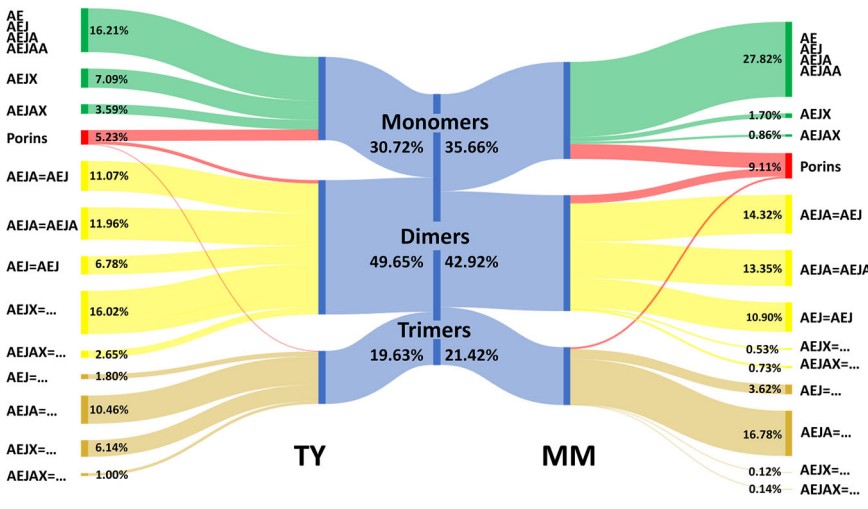

"scars" in MM may, therefore, be indicative of an increase in envelope stress. Further studying the dynamics of this process and establishing if distinct β-barrel proteins are preferentially anchored under different conditions, as has been shown in *Coxiella burnetii*[19], would provide a valuable insight into rhizobial adaptation and symbiosis. An increase in 3-3 cross-linking by L,D-transpeptidases has also previously been implicated in stress resistance and cell envelope homoeostasis[30]. In the case of *Rhizobium*, growth in MM has a significant (albeit subtle) impact on the abundance of 3-3 cross-links, but a dramatic impact on the abundance of gm-AEJX muropeptides. Looking at these specific L,D-transpeptidation products will be useful for better understanding the role that individual L,D-transpeptidases play in PG remodelling and how they contribute to cellular fitness. Finally, we demonstrated that our PG analysis strategy can uncover unexpected muropeptides like those containing unusual amino acids in the fifth position of their peptide stem (gm-AEJAX in Tables 3, 16.6% of total muropeptides). The biological significance of these unusual stems and the enzymes responsible remain unknown, but warrant investigation, as unusual residues in the fifth position are likely to impact PBP-mediated PG polymerisation.

Overall, the granularity of the PG analysis described by this work makes it possible to monitor minor changes in PG structure and composition like never before and transforms how we study the bacterial cell wall and the role it plays in helping species like *R. leguminosarum* thrive in a highly dynamic environment.

## Methods
### Bacterial strains and growth conditions
*R. leguminosarum* bv. *viciae* strain 3841[31] was grown at 28 °C in TY (5 g/L Tryptone + 3 g/L Yeast Extract + 1.3 g/L CaCl₂.6H₂O) broth or agar (15 g/L). The recipe for minimal medium is described in ref. 31. Liquid cultures were grown in 2 L flasks under agitation (180 rpm).

### PG extraction and muropeptide preparation
Cells corresponding to 500 mL of culture spun and resuspended in 20 mL of boiling Milli-Q water prior to the addition of SDS 5% (w/v) final. After 30 min at 100 °C, PG was recovered by centrifugation (2 h at 125,000 x *g*, room temperature) and washed three times in Milli-Q water. Samples were treated with trypsin (100 µg/mL) for 4 h at 37 °C in 50 mM Tris-HCl (pH 7.5). Trypsin was heat-inactivated (10 min at 65 °C) and removed by washes in Milli-Q water. The material was freeze-dried and resuspended at a concentration of 10 mg/mL.

### LC-MS/MS analysis
2 mg of purified PG was digested for 16 h using 250 units of mutanolysin (Sigma-Aldrich) in 20 mM phosphate buffer (pH 5.5) in a final volume of 200 µL. Following heat inactivation (5 min at 100 °C), the soluble

disaccharide peptides were mixed with an equal volume of 250 mM borate buffer (pH 9.25) and reduced via the addition of 25 µL of a sodium borohydride solution at 25 mg/mL. After 20 min at room temperature, the pH was adjusted to 5.0 using phosphoric acid. An Ultimate 3000 Ultra High-Performance Chromatography (UHPLC; Dionex/Thermo Fisher Scientific) system coupled with a high-resolution Q Exactive Focus mass spectrometer (Thermo Fisher Scientific) was used for LC-MS/MS analysis. Muropeptides were separated using a C18 analytical column (Hypersil Gold aQ, 1.9 µm particles, 150 × 2.1 mm; Thermo Fisher Scientific), at a temperature of 50 °C. Muropeptide elution was performed by applying a mixture of solvent A (water, 0.1% (v/v) formic acid) and solvent B (acetonitrile, 0.1% (v/v) formic acid). Following a 10 µL sample injection, MS/MS spectra were recorded over a 40 min gradient: 0–12.5% B for 25 min; 12.5–20% B for 5 min; held at 20% B for 5 min, followed by column re-equilibration for 10 min under the initial conditions. The Q Exactive Focus was operated under electrospray ionisation (H-ESI II) in positive mode. Full scan (*m/z* 150–2250) used resolution 70,000 (FWHM) at *m/z* 200, with an automatic gain control (AGC) target of $1 \times 10^6$ ions and an automated maximum ion injection time (IT). MS/MS spectra were recorded in "Top 3" data-dependent mode using the following parameters: resolution 17,500; AGC $1 \times 10^5$ ions, maximum IT 50 ms, NCE 25%, and a dynamic exclusion time of 5 seconds.

### Determination of glycan chain length and cross-linking index
Cross-linking index and glycan chain length were calculated based on the formulae described previously[32]. The cross-linking was calculated as:

$$\frac{1}{2}\left(\%\ of\ all\ dimers\right) + \frac{2}{3}\left(\%\ of\ all\ trimers\right)$$

No glycosidically-linked multimers were identified, so all dimers and trimers included in this calculation were peptide cross-linked.

Glycan chain length was inferred from the abundance of anhydroMurNAc groups, which are found at the ends of glycan chains:

$$\frac{100}{\left(\%\ of\ anhydro\ monomers\right) + \frac{1}{2}\left(\%\ of\ anhydro\ dimers\right) + \frac{1}{3}\left(\%\ of\ anhydro\ trimers\right)}$$

Because no di-anhydro muropeptides were included in the search process, they have also been excluded from the formula above.

### Byos® searches
Unbiased searches were performed using Byonic v5.2.5. For monomer searches, a FASTA file containing each peptide stem was used, and glycan moieties (gm, 480.1955 Da) were added as N-terminal modifications. For PG-anchored proteins, searches were performed against the entire *R.*

*leguminosarum* proteome. Modified peptides with a mass of 852.3600 Da (gm-AEJ) permitted once per peptide on any residue within the peptide were searched using non-specific cleavage parameters. Precursor mass tolerance was set at 8 ppm and fragment mass tolerance was set to 20 ppm for HCD fragmentation. Spectra corresponding to peptides containing an N-terminal disaccharide-tripeptide were examined manually.

## Reporting summary
Further information on research design is available in the Nature Portfolio Reporting Summary linked to this article.

## Data availability
LC-MS/MS datasets have been deposited in the GLYCOPOST repository (GPST000405). All databases and search outputs are available in Files S1–S6.

## Code availability
Code for the latest version of PGFinder can be found here: https://github.com/Mesnage-Org/pgfinder. The exact version described in this manuscript is archived here: https://github.com/Mesnage-Org/pgfinder/releases/tag/v1.3.2-ncc; https://doi.org/10.5281/zenodo.14946462.

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

## Acknowledgements
This work was funded by a BBSRC grant (BB/W013800/1 to S.M. and P.S.P.). M.G.A.Z. is funded by a Mexican government PhD scholarship (CONAHCYT, 2021-000007-01EXTF-00221). B.J.R. is the recipient of a NERC PhD iCASE studentship (NERC ACCE NE/S00713X/1). AR and IGB were supported by grants ANR Peptimet (ANR-18-CE15-0018) and IntraBacWall (ANR-16-IFEC-0004).

## Author contributions
Conceptualisation: M.G.A.Z., B.J.R., M.B., S.M.; methodology: M.G.A.Z., B.J.R., N.S., C.A.E., R.D.T., S.M.; software: B.J.R., A.V.P., N.S, R.D.T.;

validation: M.G.A.Z., S.M.; formal analysis: M.G.A.Z., B.J.R., S.M.; investigation: M.G.A.Z., B.J.R., R.L., C.A.E., A.R.; resources: R.L., P.S.P.; data curation: M.G.A.Z., C.A.E., B.J.R., N.S., R.D.T., S.M.; writing—original draft: M.G.A.Z., B.J.R., S.M.; writing—review & editing: M.G.A.Z., R.L., R.D.T., B.J.R., M.B., S.M.; visualisation: M.G.A.Z., B.J.R., S.M.; supervision: N.S., R.D.T., M.J.D., I.G.B., P.S.P., M.B., S.M.; project administration: R.D.T., M.J.D., I.G.B., P.S.P., M.B., S.M.; funding acquisition: P.S.P., S.M.

## Competing interests

The authors declare no competing interests.
