## [Transparent Peer Review file · Communications Chemistry]

A software tool and strategy for peptidoglycomics, the high-resolution analysis of bacterial peptidoglycans via LC-MS/MS

Corresponding Author: Dr Stéphane Mesnage

Version 0:

Reviewer comments:

Reviewer #1

(Remarks to the Author)

In this work, Mesnage and coworkers presented a new software tool that assists with peptidoglycomics analysis, which is a key bottleneck in the bacteriology- and microbiome-related research field. The ability to accurately and facilely deconvolute the LCMS spectra of bacterial peptidoglycan is highly desirable.

This paper developed a website interface platform for bacterial peptidoglycan (PG) analysis via LC-MS/MS. First, they proposed a PGLang to represent complex and various structures of muropeptides. Second, they illustrated the procedures to analyze the LC-MS/MS raw data of *Rhizobium leguminosarum* PG stepwise, from monomers to modifications, to MS/MS analysis, multimers, and even porin anchored to PG. Finally, the authors compared muropeptide compositions of *R. leguminosarum* in minimal media (MM) and rich media (TY). This study is well-written and shows new merits. The use of the website interface for PG analysis reduced the hardware requirements for user computers to load and calculate, which is an accessible way to generate the fragment prediction. In addition, the porin analysis is quite novel. However, I do have some questions and comments for the authors to improve the work.

1. The incorporation of PGLang is useful for the systematic naming of PG and for annotating the MS/MS fragments. However, it is unclear how the list of PGLang can be easily created by the user in this study. Please elaborate on this building process. Are customized databases needed for different bacteria?
2. In addition, the authors can improve PGLang with the implementation to convert to SMILEs, which is more general and informative, including the chemical drawing, calculation of molecular properties, isotope ratio etc. This will be more understandable to someone outside the PG field to do the analysis.
3. In Figure 1, please include the chemical structure of muropeptides to demonstrate the complexity of PG and the necessity of introducing PGLang. Besides, please include the canonical structure of bacterial PG as background knowledge in the paragraph on Page 6 (line 121) so that readers unfamiliar with PG can better understand the content.
4. I find the criticism of the reporting of m/z of reduced 940 [401-402] is splitting hairs – the m/z reported in previous papers could be explained by authors: (1) reporting the experimental value; (2) rounding the value due to mass accuracy of their instrument; (3) were lazy and did not check the convention for reporting m/z. In any case, reporting the m/z to six d.p. is trivial and anybody can do it if they had the chemical formula. The more consistent way to share these m/z values would be to always include the chemical formula so others can calculate the m/z to whatever accuracy they require.
5. While the authors commented that previous work, such as PGN_MS2, uses external software for MS/MS comparison, this study also used external software Byonic™ to analyze MS/MS spectra. If I understand it correctly, the authors generate in silico fragment lists for each PGN and then match them against MS/MS spectra manually. The manual way of analyzing the MS/MS spectra sounds very tedious and less efficient. Please comment.
6. Can the relative intensity of MS/MS fragment peaks can be predicted in the software?
7. While the writing is clear, the figures are rather off-putting. The authors should greatly simplify Fig. 5, 6, 7, and 8 for the ease of reading. Panels that are just screen captures should be redrawn when possible, as they include words that cannot be read at that resolution.

Reviewer #2

(Remarks to the Author)

This manuscript presents a new informatics toolset to analyse mass spectrometry data for the identification of mucopeptides that are central in the peptidoglycan structure. This is an original and valuable piece of work but poorly reported. The text is overly descriptive and not analytical enough. It is heterogeneous, focusing on details or making high level statements with weak transitions and it confuses the reader in the process. A number of sales pitch like superlatives and self centred expressions should be removed.

The challenge of analysing the peptidoglycan intricate molecular structure is real and justifiably requires standards, reference datasets and appropriate representations. Only with these in place makes automation possible. Despite the authors' acute awareness of this situation, the manuscript is shaped as a mix of introducing new standard(s), tutoring software usage and validating a method. In the end, it defeats its intended didactic purpose. This situation is also reflected in the (too) many figures needed to illustrate some points.

Now, the content is there but the presentation requires substantial reshaping and a clear set of messages to pass on to the readers.

Examples of issues in the Introduction

The topic is well introduced at first but needs to be polished in the 2nd half.

page 3, line 54: the transition from general to specific is abrupt. It misses a simple "Peptidoglycan also plays a role in symbiosis. For example, it was shown that..."

page 3-4, line 73-78: this level of detail should be moved to the discussion, keeping less descriptive statements for the introduction

page 4, line 82: what does "to reduce the barrier to entry associated with PG structural analysis" mean?

page 4, line 84: "We standardize a novel, intuitive, and universal language..." is rather presumptuous. A standard has to be used before becoming one. "We propose to standardize..." would sound more to the point. Then in the same sentence "...and a tool" suggests that standardisation applies to that as well.

page 4, line 84-92: this list itemising the different parts of the work reflect the lack of structure of the manuscript. It is a succession of achieved tasks as opposed to conveying key messages on achievements. What does this work hinge on?

Results

Headers again reveal the issues with the presentation:

1) "Novel software tools for the structural analysis of peptidoglycan via LC-MS/MS"

spans the definition of a standard new representation and a new interface. It hardly refers to software per se.

2) "Rhizobium leguminosarum as a model system for exploring peptidoglycan structure and remodelling" is a tutorial on how to use the toolset. It turns out it depends on four steps that are not even mentioned from the beginning. Why not use this structure to explain the methodology earlier and expand from there?

3) "Applying our pipeline to quantify changes in PG composition between growth conditions", this validation part is essential and probably the best explained. Figure 8 should in fact be the basis of a much earlier figure that highlights the various aspects of the analysis.

Discussion

It is likely to be rewritten if the Results section is redone. The authors have to improve the outline of the discussion that jumps from the need for standards to the quality of the results from a biological point of view and hardly elaborates upon to the technical added value of their solution.

Minor:

page 6, line 123: "model"S

The possessive case is often misused. A piece of software does not possess anything.

page 7, line 134: "for the sake of" usually applies to something not people unless turned into a possessive case...

Reviewer #3

(Remarks to the Author)

The authors describe new software tools to perform the analysis of LC-MS/MS of bacterial peptidoglycans. To prove their concept, they used a model bacteria *R. leguminosarum* grown in a rich or minimal media. Then they step by step they describe how the analysis is performed: 1. To use PG monomers using MS and MS/MS, 2. To identify peptidoglycan modifications, 3. To identify outer membrane proteins covalently anchored to the PG, 4. To identify PG multimers and confirming their structure. They apply the software to compare the PG composition in the two different growth media. The manuscript is easy to follow and can be understood well by scientists not in the field. The manuscript appears as an application with a different bacterium from the author eLife previous manuscript with incremental advances.

For example, in the current manuscript, a new language is claimed for the PG peptides PGlang. The peptides are simplified with letter codes. While in the previous manuscript, the peptides were described as their long form. Which was definitely easier for the reader in the previous elife manuscript. There is a need for a new language code in this field but it is time consuming for the reader to integrate and go back to figure 1 to really understand.

From the previous eLife manuscript, similar Sankey plots are produced such as Figure 9. Another downside to the manuscript is that it aims to describe new software tools. However, they use an existing software Bionic. What they actually do is to expand Bionics to their workflow which is different than to develop new software tools. It would be important to clarify: 1. What is different from their previous manuscript (new databases?), 2. To Clarify what they claim as new software tools (new interface?(Figure 3.)) 3. Comment on how more than two different conditions can be compared meaningfully with the interface?

Major clarifications are needed in term of why it is important to study bacterial peptidoglycan by LC-MS. If we look at disruption of the peptidoglycan layer by antibiotics for example, the study is usually simple and less time consuming than running LC-MS, with bacteria being tested in minimum inhibitory concentration tests that are not time consuming. Precision on what this cross-linking index is, and how is it calculated would be useful for the reader. How is glycan chain length determined, considering that we are speaking about LC/MS/MS and usually fragments are observed is also an important aspect to describe.

Minor changes:

Figure 2. A and B have formatting mistakes in them when printed

Picture 6B is of low quality I can barely see what ions are in the MS. Picture 7C is also not easy to read and very blurry same with Figure 8.

Some sentences in the manuscript are unprecise and could benefit from rewording, for example:

p.20 line 395 "Things get even more confusing"

p.20 line 394 "some papers even refer to"

p.21 line 414 "manually check the fragmentation spectra"

p.21 line 419 "an incomplete description of its dependencies" In this case, which dependencies causes the issues?

p.21 line 427 "Any service would do"

p.431 line 431 "cutting out"

p.4 line 79 "Vendor-neutral identifications and quantifications of muropeptides"

Version 1:

Reviewer comments:

Reviewer #1

(Remarks to the Author)

Remarks to the Author

Concerns and questions raised in comments 1-4,6 have been clarified by the authors. Further suggestions for comment 5/7 are given below. The authors have added a short description of Glauner's methods to determine glycan chain length and crosslinking index in response to another reviewer which is slightly inaccurate:

- the cross-linking index should exclude glycosidic-linked trimers
- the formula for chain length given assumes only one anhydroMurNAc modification is present

Comment 5: I agree with the author that this is beyond the scope of the work. I find that lines 418-419 give the wrong impression that PGfinder can analyse raw MS data files directly. If I am not mistaken, raw MS data files must first be processed to extract features (Figure 1).

Comment 7: The new figures are a big improvement to the original. However, Figure 5 is still too complex in my opinion. For comparison, the flowchart in Figure 2 in their 2021 paper is simpler to read. Given that this work is an expansion to their 2021 work, similar colour schemes can be employed. I note that the title of the work has been changed to emphasise "four steps" yet Figure 5 shows five steps. Based on my interpretation, the steps to generate DB_3 and DB_4 should be combined into a single step that generates modifications from DB_2 (as explained in the text). The screenshots (TICs, tables) are unnecessary distractions and are best removed. Also, I suggest adding a short name to each DB (in the figure) to clarify their purposes:

DB_1: monomer scan

DB_2: detected monomers

DB_3/4: modified monomers

DB_5: final

Reviewer #2

(Remarks to the Author)

The manuscript has greatly benefited from accounting for all reviewers' comments. My concerns have been addressed and I maintain my earlier opinion on the originality of this work. This revised version should be published.

Reviewer #3

(Remarks to the Author)

I am satisfied with the changes brought to the manuscript.

Reviewer #1 (Remarks to the Author):

In this work, Mesnage and coworkers presented a new software tool that assists with peptidoglycomics analysis, which is a key bottleneck in the bacteriology- and microbiome-related research field. The ability to accurately and facilely deconvolute the LCMS spectra of bacterial peptidoglycan is highly desirable.

This paper developed a website interface platform for bacterial peptidoglycan (PG) analysis via LC-MS/MS. First, they proposed a PGLang to represent complex and various structures of muuropeptides. Second, they illustrated the procedures to analyze the LC-MS/MS raw data of *Rhizobium leguminosarum* PG stepwise, from monomers to modifications, to MS/MS analysis, multimers, and even porin anchored to PG. Finally, the authors compared muuropeptide compositions of *R. leguminosarum* in minimal media (MM) and rich media (TY). This study is well-written and shows new merits. The use of the website interface for PG analysis reduced the hardware requirements for user computers to load and calculate, which is an accessible way to generate the fragment prediction. In addition, the porin analysis is quite novel. However, I do have some questions and comments for the authors to improve the work.

1. The incorporation of PGLang is useful for the systematic naming of PG and for annotating the MS/MS fragments. However, it is unclear how the list of PGLang can be easily created by the user in this study. Please elaborate on this building process. Are customized databases needed for different bacteria?

Currently, databases are constructed by hand in a text editor, though we have provided basic databases for model organisms (*E. coli*, *B. subtilis*, *S. aureus*, etc). We do anticipate that customized databases will be most frequently needed since the search space depends on the scope of the analysis and the type of peptidoglycan analysed. We have added a comment in the text to clarify this (p. 9, L. 171–173).

2. In addition, the authors can improve PGLang with the implementation to convert to SMILES, which is more general and informative, including the chemical drawing, calculation of molecular properties, isotope ratio etc. This will be more understandable to someone outside the PG field to do the analysis.

We thank the reviewer for this suggestion. We have now added this capability to the mass calculator module. Each database built from a list of structures in PGLang (.txt) now returns a monoisotopic mass and SMILES. This is now indicated in the results (p. 7, L. 148–150) and the discussion (p. 21, L. 436–441), and was used to automatically generate the chemical drawing in Fig. 2A.

3. In Figure 1, please include the chemical structure of muuropeptides to demonstrate the complexity of PG and the necessity of introducing PGLang. Besides, please include the canonical structure of bacterial PG as background knowledge in the paragraph on Page 6 (line 121) so that readers unfamiliar with PG can better understand the content.

For space reasons we've included only a full chemical drawing of a monomeric muuropeptide, but one with both branches and modifications in Fig. 2A. Fig. 2 therefore shows how the full chemical structure relates to the graphical shorthand that we use to show more complex, multimeric muuropeptide structures in Fig. 3. Background knowledge on the canonical structure of PG was introduced (p. 6, L. 120–123).

4. I find the criticism of the reporting of m/z of reduced 940 [401-402] is splitting hairs – the m/z reported in previous papers could be explained by authors: (1) reporting the experimental value; (2) rounding the value due to mass accuracy of their instrument; (3) were lazy and did not check the convention for reporting m/z . In any case, reporting the m/z to six d.p. is trivial and anybody can do it if they had the chemical formula. The more consistent way to share these m/z values would be to always include the chemical formula so others can calculate the m/z to whatever accuracy they require.

The inconsistency of masses deals with **theoretical** monoisotopic masses, not observed masses. We have added this adjective to avoid confusion. We fully agree that calculating theoretical monoisotopic masses (or m/z values) is trivial, however this is clearly not achieved in the field. We argue that PGLang is a much easier way to describe molecules than chemical formula, which is more prone to human error — particularly when dealing with modification and bond losses / gains. Our tool solves this problem, and with the newly-added conversion to SMILES, a chemical formula can also be trivially computed using existing tools. The sentences (p.21, L. 432–438) have been clarified.

5. While the authors commented that previous work, such as PGN_MS2, uses external software for MS/MS comparison, this study also used external software Byonic™ to analyze MS/MS spectra. If I understand it correctly, the authors generate in silico fragment lists for each PGN and then match them against MS/MS spectra manually. The manual way of analyzing the MS/MS spectra sounds very tedious and less efficient. Please comment. This is a fair point. We have used Byonic™ for monomers but we used our fragment predictor for all dimers (since Byonic is incapable of handling these). We have been transparent about the current limitation of our software and have added more nuance to the PGN_MS2 comparison — making clear that whilst we automate more of the search process than PGN_MS2 does (by automating MS1), we have further to go and desire to automate MS/MS analysis as well in the future. Unfortunately, this is not a straightforward task, so we feel that it is beyond the scope of this work (p21, L. 418–423).

6. Can the relative intensity of MS/MS fragment peaks can be predicted in the software? Intensities are currently not predicted. This is in theory possible, but relative intensities of MS/MS fragments are partly dependent on unstandardized fragmentation settings of the instrument and provide comparatively little information versus presence or absence.

7. While the writing is clear, the figures are rather off-putting. The authors should greatly simplify Fig. 5, 6, 7, and 8 for the ease of reading. Panels that are just screen captures should be redrawn, when possible, as they include words that cannot be read at that resolution. We have simplified figures 5, 7, 8 and checked their resolution. Figure 6 illustrated the results provided in Table 2 and was therefore redundant; it has now been moved to Supplementary information (Fig. S5).

Figure 5

Figure 7

Figure 8

Reviewer #2:

This manuscript presents a new informatics toolset to analyse mass spectrometry data for the identification of muropeptides that are central in the peptidoglycan structure. This is an original and valuable piece of work but poorly reported. The text is overly descriptive and not analytical enough. It is heterogeneous, focusing on details or making high level statements with weak transitions and it confuses the reader in the process. A number of sales pitches like superlatives and self-centred expressions should be removed.

The challenge of analysing the peptidoglycan intricate molecular structure is real and justifiably requires standards, reference datasets and appropriate representations. Only with these in place makes automation possible.

Despite the authors' acute awareness of this situation, the manuscript is shaped as a mix of introducing new standard(s), tutoring software usage and validating a method. In the end, it defeats its intended didactic purpose. This situation is also reflected in the (too) many figures needed to illustrate some points.

Now, the content is there but the presentation requires substantial reshaping and a clear set of messages to pass on to the readers.

We agree that the original manuscript contained many complex figures and supplementary materials. We hope that the simplified revised version is easier to follow.

The introduction, software results, and discussion have been reorganised or rewritten, making clear that the work was motivated by a desire to improve on PGFinder's prior shortcomings on the road to an eventual one-click analysis tool. The introduction of PGLang has been reframed as an essential pre-requisite to the development of the new Mass Calculator and Fragment Generator modules, better incorporating it into the rest of the manuscript.

As for the "...four-step PG analysis strategy" section, we believe that the level of detail provided, and the comprehensive description of the analysis strategy addresses a gap in the original PGFinder paper and better puts the tool into context of a complete LC-MS/MS analysis pipeline (p.21, L. 445–447). Over the past 5 years, the corresponding author's lab has hosted and trained several students and postdocs for PG analysis. Based on regular interactions with groups keen to learn how to perform PG analysis, we think that this level of detail is required for users to understand how the tools work and how to use them.

Examples of issues in the Introduction

The topic is well introduced at first but needs to be polished in the 2nd half.

page 3, line 54: the transition from general to specific is abrupt. It misses a simple "Peptidoglycan also plays a role in symbiosis. For example, it was shown that..."

A transition has been added to better link general and specific. (p.3, L54-58)

page 3-4, line 73-78: this level of detail should be moved to the discussion, keeping less descriptive statements for the introduction.

The text has been modified as suggested (the text was moved).

page 4, line 82: what does "to reduce the barrier to entry associated with PG structural analysis" mean?

References to the "reduced barrier to entry" have been replaced with "increased ease-of-use". (p.4, L78 and p.20, L398).

page 4, line 84: "We standardize a novel, intuitive, and universal language..." is rather presumptuous. A standard has to be used before becoming one. "We propose to standardize..." would sound more to the point. Then in the same sentence "...and a tool" suggests that standardisation applies to that as well.

The text now refers to PGLang as a "formal language" — it has a syntax standard that's specified using a formal grammar (EBNF), but we agree that having a written standard doesn't make something standard in the conventional sense. We have also more clearly linked PGLang to the software tools that it enables.

page 4, line 84-92: this list itemising the different parts of the work reflect the lack of structure of the manuscript. It is a succession of achieved tasks as opposed to conveying key messages on achievements. What does this work hinge on?

The rewritten introduction hopefully makes it clearer that the manuscript has been reorganised around the improvements to PGFinder. A few highlights from the software and biology results are still included, but it has been made much more explicit what the key takeaways are and how they are related to / rely on one another.

Results

Headers again reveal the issues with the presentation:

1) "Novel software tools for the structural analysis of peptidoglycan via LC-MS/MS" spans the definition of a standard new representation and a new interface. It hardly refers to software per se.

In hindsight, we agree that this section was hard to follow. The text has been rewritten as three sections (foreshadowed by the new introduction, with PGLang being the key enabler of the newly added modules):

- Enhancing PGFinder's existing functionality with an improved MS1 output and web interface
- Condensing complex mucopeptide structures into PGLang, a concise formal language
- Expanding PGFinder to automate mass calculation and MS/MS fragment prediction

2) "*Rhizobium leguminosarum* as a model system for exploring peptidoglycan structure and remodelling" is a tutorial on how to use the toolset. It turns out it depends on four steps that are not even mentioned from the beginning. Why not use this structure to explain the methodology earlier and expand from there?

We have now described the search strategy up front and included the simplified Figure 8 at this stage of the manuscript (Figure 5 in the revised manuscript). The title has been changed to reflect this change (***Rhizobium leguminosarum* as a model system for describing a four-step PG analysis strategy**).

3) "Applying our pipeline to quantify changes in PG composition between growth conditions", this validation part is essential and probably the best explained. Figure 8 should in fact be the basis of a much earlier figure that highlights the various aspects of the analysis.

Figure 8 has been moved much earlier in the revised manuscript as suggested by the reviewer, it is now Figure 5 (see comment above).

Discussion

It is likely to be rewritten if the Results section is redone. The authors have to improve the outline of the discussion that jumps from the need for standards to the quality of the results from a biological point of view and hardly elaborates upon to the technical added value of their solution.

The discussion has been rewritten, now with the software and biology more clearly separated. A new paragraph dedicated to the technical comparison of PGFinder, HAMA, and PGN_MS2 has been included immediately after summarizing the key software takeaways (p.20–21, L. 396–423).

Minor:

page 6, line 123: "model"S

The possessive case is often misused. A piece of software does not possess anything.

This text was naturally removed during rewriting — p.20–21, L. 415–418 is where the content ended up, though phrased differently.

page 7, line 134: "for the sake of" usually applies to something not people unless turned into a possessive case...

Removed during rewriting.

Reviewer #3 (Remarks to the Author):

The authors describe new software tools to perform the analysis of LC-MS/MS of bacterial peptidoglycans. To prove their concept, they used a model bacteria *R. leguminosarum* grown in a rich or minimal media. Then they step by step they describe how the analysis is performed: 1. To use PG monomers using MS and MS/MS, 2. To identify peptidoglycan modifications, 3. To identify outer membrane proteins covalently anchored to the PG, 4. To identify PG multimers and confirming their structure. They apply the software to compare the PG composition in the two different growth media. The manuscript is easy to follow and can be understood well by scientists not in the field. The manuscript appears as an application with a different bacterium from the author eLife previous manuscript with incremental advances.

For example, in the current manuscript, a new language is claimed for the PG peptides PGlang. The peptides are simplified with letter codes. While in the previous manuscript, the peptides were described as their long form. Which was definitely easier for the reader in the previous eLife manuscript. There is a need for a new language code in this field but it is time consuming for the reader to integrate and go back to figure 1 to really understand.

From the previous eLife manuscript, similar Sankey plots are produced such as Figure 9. Another downside to the manuscript is that it aims to describe new software tools. However, they use an existing software Bionic. What they actually do is to expand Bionics to their workflow which is different than to develop new software tools.

To help justify learning this new language, it's been made clearer the key role it plays in the new Mass Calculator and Fragment Predictor tools. Additionally, the software now also translates this simplified language to SMILES if atomic detail is desired. New languages always take effort to learn, but we believe ours is one of the more intuitive and less ambiguous options (see p.21, L. 425–429).

Our software is capable of several things (like predicting the fragmentation of branched or multimeric) muuropeptides that Byonic is not and can be used completely Byonic-free. We use Byonic to supplement a workflow largely driven by our own tools whilst we work on integrating MS2 analysis into PGFinder itself. In the same way that PGN_MS2 was a new software tool (despite relying on MS-DIAL for any form of MS analysis), we believe our Fragment Generator and Mass Calculator modules are novel developments.

It would be important to clarify:

1. What is different from their previous manuscript (new databases?)

The introduction and software results sections have been rewritten to make it clearer that most of the novelty comes from the improvements to PGFinder, with the remaining contribution being the detailed description of our complete LC-MS/MS pipeline (which the previous manuscript didn't describe in detail).

2. To Clarify what they claim as new software tools (new interface?(Figure 3.))

Both the introduction and software results sections are now much more explicit about the software improvements. The introduction sorts them into two key areas (p.4, L. 75–77), then one new ("Improving PGFinder's existing functionality...") and one greatly expanded results section ("Expanding PGFinder to automate...") go into further detail.

3. Comment on how more than two different conditions can be compared meaningfully with the interface.

The interface and tool are used to generate the muuropeptide quantifications that can then be used in statistical comparisons (added p.10, L. 193) outside of PGFinder. Though we'd like to automate this process in the future, PGFinder do not currently perform any statistical comparisons itself.

Major clarifications are needed in term of why it is important to study bacterial peptidoglycan by LC-MS. If we look at disruption of the peptidoglycan layer by antibiotics for example, the study is usually simple and less time consuming than running LC-MS, with bacteria being

tested in minimum inhibitory concentration tests that are not time consuming. Precision on what this cross-linking index is, and how is it calculated would be useful for the reader. How is glycan chain length determined, considering that we are speaking about LC/MS/MS and usually fragments are observed is also an important aspect to describe.

A paragraph of the rewritten introduction has been dedicated to justifying the use of LC-MS/MS (p.3, L. 59–70). In our particular case, it's essential for tracking changes in low-abundance muropeptides that result from the activity of L,D-transpeptidases (as they exchange amino acids in the fourth-residue position of peptide stems and/or covalently anchor proteins to the peptidoglycan). This has also been more explicitly spelled out in the rewritten discussion (p.22, L. 452–476).

A new section has been added to the methods that describes the calculation of cross-linking index and glycan chain length (p.25, L. 517–522).

Minor changes:

Figure 2. A and B have formatting mistakes in them when printed

Figures have been revised and exported as PNGs, which should display the same everywhere.

Picture 6B is of low quality I can barely see what ions are in the MS.

Figure moved to supplementary (S5) and quality improved

Picture 7C is also not easy to read and very blurry same with Figure 8.

The issues with figure resolution seem to be a result of Word compressing embedded images. We disabled this and have checked the resolution of all main figures submitted as individual PNG files and included supplementary figures as original Microsoft Office files.

Some sentences in the manuscript are unprecise and could benefit from rewording, for example:

p.20 line 394 “some papers even refer to”

The text has been changed (p.21, L. 429)

p.20 line 395 “Things get even more confusing”

The text has been changed (p.21, L. 429–430)

p.21 line 414 “manually check the fragmentation spectra”

Text removed

p.21 line 419 “an incomplete description of its dependencies” In this case, which dependencies causes the issues?

The text has been changed (p.20, L. 409–410) — it's the Python dependencies, but too many to neatly describe in the manuscript. Another researcher has found the same issue and submitted a pull request: https://github.com/jerickwan/PGN_MS2/pull/1/files

p.21 line 427 “Any service would do”

Text removed

p.431 line 431 “cutting out”

Text removed

p.4 line 79 “Vendor-neutral identifications and quantifications of muropeptides”

Text removed

Reviewer #1 (Remarks to the Author):

Concerns and questions raised in comments 1-4,6 have been clarified by the authors. Further suggestions for comment 5/7 are given below.

The authors have added a short description of Glauner's methods to determine glycan chain length and crosslinking index in response to another reviewer which is slightly inaccurate:

- the cross-linking index should exclude glycosidic-linked trimers
- the formula for chain length given assumes only one anhydroMurNAc modification is present

It's been clarified in the text (p.25- L509--518). Because we did not identify any glycosidic-linked multimers (Fig. S3E), there is no need to exclude them from the cross-linking index calculation. We have also clarified in the text that we're using a simplified version of the Glauner equation for glycan chain length that excludes di-anhydro dimers and trimers, since our search process did not include these muropeptides.

Comment 5: I agree with the author that this is beyond the scope of the work. I find that lines 418-419 give the wrong impression that PGFinder can analyse raw MS data files directly. If I am not mistaken, raw MS data files must first be processed to extract features (Figure 1).

The reviewer's understanding is correct. The text has been changed to make this point clearer; "eliminating the need for a separate tool" was replaced by "only requiring additional software for data deconvolution / feature extraction" (p.21, L411).

Comment 7: The new figures are a big improvement to the original. However, Figure 5 is still too complex in my opinion. For comparison, the flowchart in Figure 2 in their 2021 paper is simpler to read. Given that this work is an expansion to their 2021 work, similar colour schemes can be employed.

In our 2021 work, the colours were used to demarcate the four different "steps" of our software's pipeline. Here we found it clearer to organise our five analysis steps into labelled columns instead of requiring the reader to reference the legend for a colour key.

I note that the title of the work has been changed to emphasise "four steps" yet Figure 5 shows five steps. Based on my interpretation, the steps to generate DB_3 and DB_4 should be combined into a single step that generates modifications from DB_2 (as explained in the text).

We found it simpler to keep the overall step structure unchanged, but to refer to the strategy as five-step instead of four-step. This change has been reflected in Figure 5 ("FINAL STEP" has been replaced by "STEP 5"). The text has been modified accordingly to reflect this change:

- A reference to 5 steps is also in the text (p.9, L168; p.10, L184)
- the section heading "Applying our pipeline..." is now "Step 5: Final quantification of muropeptides and comparison of growth conditions" (p.17, L339)

The screenshots (TICs, tables) are unnecessary distractions and are best removed.

ALL screenshots have been removed as suggested.

Also, I suggest adding a short name to each DB (in the figure) to clarify their purposes:

DB_1: monomer scan

DB_2: detected monomers

DB_3/4: modified monomers

DB_5: final

Databases have been given short names as suggested; the revised figure is shown below.

Reviewer #2 (Remarks to the Author):

The manuscript has greatly benefited from accounting for all reviewers' comments. My concerns have been addressed and I maintain my earlier opinion on the originality of this work. This revised version should be published.

No modifications requested.

Reviewer #3 (Remarks to the Author):

I am satisfied with the changes brought to the manuscript.

No modifications requested.